# Structural Heterogeneity of the GABAergic Tripartite Synapse

**DOI:** 10.3390/cells11193150

**Published:** 2022-10-07

**Authors:** Cindy Brunskine, Stefan Passlick, Christian Henneberger

**Affiliations:** 1Institute of Cellular Neurosciences, Medical Faculty, University of Bonn, 53127 Bonn, Germany; 2German Center for Neurodegenerative Diseases (DZNE), 53127 Bonn, Germany

**Keywords:** astrocytes, morphology, perisynaptic astrocytic processes, GABA, inhibitory synapses

## Abstract

The concept of the tripartite synapse describes the close interaction of pre- and postsynaptic elements and the surrounding astrocyte processes. For glutamatergic synapses, it is established that the presence of astrocytic processes and their structural arrangements varies considerably between and within brain regions and between synapses of the same neuron. In contrast, less is known about the organization of astrocytic processes at GABAergic synapses although bi-directional signaling is known to exist at these synapses too. Therefore, we established super-resolution expansion microscopy of GABAergic synapses and nearby astrocytic processes in the *stratum radiatum* of the mouse hippocampal CA1 region. By visualizing the presynaptic vesicular GABA transporter and the postsynaptic clustering protein gephyrin, we documented the subsynaptic heterogeneity of GABAergic synaptic contacts. We then compared the volume distribution of astrocytic processes near GABAergic synapses between individual synapses and with glutamatergic synapses. We made two novel observations. First, astrocytic processes were more abundant at the GABAergic synapses with large postsynaptic gephyrin clusters. Second, astrocytic processes were less abundant in the vicinity of GABAergic synapses compared to glutamatergic, suggesting that the latter may be selectively approached by astrocytes. Because of the GABA transporter distribution, we also speculate that this specific arrangement enables more efficient re-uptake of GABA into presynaptic terminals.

## 1. Introduction

Inhibitory synaptic transmission plays many important roles in the brain. In cortical areas such as the hippocampus, the main neurotransmitter mediating phasic inhibition is GABA, which is also involved in setting the strength of tonic inhibition. Similar to glutamatergic synapses, there are many interactions between GABAergic synapses and GABAergic signaling and astrocytes, which has been covered extensively by recent reviews [1,2]. For example, astrocytes express the GABA transporter 3 (GAT3) [3,4], which regulates, for instance, tonic GABAergic inhibition [5] and can drive astrocytic Ca^2+^ increases [6]. Astrocytes also sense the activity of GABAergic synapses via metabotropic GABA-B receptors that trigger astrocytic Ca^2+^ signals and downstream signaling cascades [7]. At the same time, astrocytes have been shown to potentiate GABAergic synaptic transmission in a Ca^2+^-dependent manner [8] and to release GABA by multiple mechanisms [9,10,11]. These examples have in common that the interaction between cell types involves the diffusion of GABA or another signaling molecule between them. Therefore, the distance and spatial arrangements of the compartments determine the efficacy of such bi-directional signaling.

The configuration of perisynaptic astrocytic processes and hippocampal glutamatergic synapses has been studied extensively [12,13,14,15,16,17]. Structural plasticity of perisynaptic astrocytic processes near hippocampal glutamatergic synapses has also been demonstrated by several studies [18,19,20,21,22,23]. It can modify the astrocytic regulation of synaptic transmission [22], the stability of glutamatergic synapses [18], and extracellular glutamate spread and synaptic crosstalk [20]. In comparison, studies investigating the structure of perisynaptic astrocytic processes near GABAergic synapses and the structural relationship between both are relatively scarce [12,24].

This is a gap in the current understanding of tripartite synapse structure because the geometry of astrocytic processes near excitatory and inhibitory synapses is likely to differ. For example, CA3-CA1 excitatory synapses onto hippocampal pyramidal cells in the *stratum radiatum* mostly target dendritic spines whereas inhibitory inputs mainly terminate on dendritic shafts [25], which puts different spatial constraints on the ability of astrocyte processes to approach the synapses. Furthermore, an astrocyte covers and interacts with both glutamatergic and GABAergic synapses in its territory. It has been speculated that this may require subcellular specialization to be physiologically meaningful [26], which could be realized on the structural level. In addition, there is the open question if within the territory of an astrocyte its processes invade and fill the neuropil non-selectively or if there are specific patterns to it, for instance related to synapse type.

For these reasons, we established super-resolution expansion microscopy [27,28] of GABAergic synapses and nearby astrocyte processes. This enabled us to then explore the structural details of GABAergic synapses, the distribution of astrocytic processes near those synapses, and to compare the latter between structurally different GABAergic synapses and with glutamatergic synapses in the same brain region, which we had studied previously [13].

## 2. Results

### 2.1. Expansion Microscopy of GABAergic Synapses and Perisynaptic Astroglia

Fine details of synapses and perisynaptic astrocytic processes cannot be fully resolved using diffraction limited microscopy. We have previously used expansion microscopy (ExM) [27,28] to visualize perisynaptic astrocytic processes and the glutamatergic synapses they approach by taking advantage of the 4–5-fold resolution increase achievable by this ExM protocol [13,29,30]. Here, we used the same approach to analyze the spatial configuration of GABAergic synapses in the *stratum radiatum* of the mouse hippocampus (CA1) and of nearby astrocytic processes. GABAergic synapses were labeled presynaptically using an antibody against the vesicular GABA transporter (VGAT), because VGAT localizes to synaptic vesicles of inhibitory terminals (GABAergic and glycinergic) [31] and glycinergic synapses are believed to be absent or very rare in the CA1 *stratum radiatum* [32]. The postsynaptic counterpart was labeled using antibodies against gephyrin, because gephyrin is an integral part and organizer of GABA and glycine receptor clusters on the postsynaptic neurons [33]. For astrocytes, the fluorescence of EGFP expressed in the cytosol of astrocytes [34] was amplified with GFP antibodies. An example of confocal microscopy of this triple-label immunohistochemistry is shown in Figure 1A. For the structural analysis of GABAergic synapses and perisynaptic astrocytic process this approach was combined with ExM. An example of a subregion of an astrocyte in the CA1 *stratum radiatum* and the GABAergic synapses embedded in its territory is shown in Figure 1B.

For comparisons with previous studies, we first analyzed the density of putative presynaptic GABAergic terminals and postsynaptic gephyrin clusters by binarizing images using a visually adjusted threshold and particle counting (6 image planes from 10 image stacks, from 4 independent experiments, and 3 animals). We observed a VGAT density of 78.4 ± 7.6 and gephyrin density of 58.2 ± 6.5 per 1000 µm^2^ (Figure 2A). This is somewhat lower than what had been reported previously (VGAT ~156 per 1000 µm^2^ and gephyrin ~112 per 1000 µm^2^) [35], which could be explained by differences in setting the fluorescence intensity threshold. However, our estimates are in good agreement with other studies in this region, which observed an about 20 to 40-fold higher density of excitatory compared to inhibitory synapses in this region [25,36] and a density of excitatory synapses of ~1500 per 1000 µm^2^ [36].

Next, we investigated the relationship between presynaptic VGAT cluster size and postsynaptic gephyrin cluster number and size. As described previously [37,38,39], postsynaptic gephyrin can be clustered in subdomains. This was readily and abundantly detectable in the present data set, where VGAT clusters were immediately adjacent to up to four visually identifiable gephyrin clusters (Figure 2B,C). Comparing the integrated VGAT and gephyrin cluster fluorescence intensities between synapses with one, two, or three gephyrin clusters, we observed no differences for VGAT and a modest increase in total gephyrin (Figure 2D,E). This is similar to findings from a previous study using three-dimensional structured illumination microscopy [39], which demonstrated that gephyrin and VGAT total volume and cluster number do not correlate.

### 2.2. Astrocytic Volume Distribution at GABAergic and Glutamatergic Synapses

We next explored how astrocytic processes are distributed around GABAergic synapses and how that distribution depends on GABAergic synapse properties. To do so, the average fluorescence intensity of EGFP, expressed in the cytosol of astrocytes, was analyzed at increasing distances from a synaptic point of reference. This quantifies how the astrocyte volume is distributed around synapses [13,15,20]. We chose this approach instead of, for instance, reconstruction of synaptic and astrocytic surfaces because that would involve the setting of a fluorescence intensity threshold, which can be difficult to standardize across preparations and experiments. As previously described [13], the average astrocytic EGFP fluorescence was determined in three-dimensional spherical shells with increasing diameter (Figure 3A) to describe the relationship between astrocytic volume and distance to the synapses.

In the first analysis, we concentrated on VGAT clusters in contact with one gephyrin cluster and centered the analysis of the astrocyte volume on the middle of the presumed synaptic cleft at the interface of VGAT and gephyrin fluorescence (Figure 3A, right panel). The EGFP profiles around 10 randomly chosen synapses per astrocyte territory were obtained and the background fluorescence was subtracted for each profile. The profiles of individual astrocytes were averaged and normalized by the fluorescence intensity observed at a distance of 0.9 to 1.0 µm. This procedure was chosen because EGFP expression levels and imaging conditions vary between astrocytes and ExM experiments but not between synapses in the territory of a single astrocyte. We found that astrocyte processes started appearing at a distance of 0.2 µm from the synapse center and that their volume reached near-plateau values at a distance of about 0.6–0.8 µm (Figure 3B, filled circles). As a control, we also analyzed the astrocyte volume distribution around randomly chosen points in the image stack (Figure 3B, empty circles), which did not show a distance dependence as expected. Next, we compared this astrocyte volume distribution at GABAergic synapses to that at glutamatergic synapses by re-analyzing a subset of ExM data from our previous study [13] in an identical manner. Interestingly, we found that the astrocytic volume increased faster with distance from the synapse compared to GABAergic synapses (Figure 3C). Because the EGFP distribution at glutamatergic synapses appeared to have a peak at a distance of ~0.6 µm, we also compared the peak-scaled profiles and obtained the same significant difference (repeated measures two-way ANOVA, distance *p* < 0.0001, interaction of distance and synapse type *p* = 0.0072). To quantify this difference further, we compared the normalized EGFP fluorescence at a distance of 0.2–0.4 µm and found that it was higher at glutamatergic synapses (glu: 0.62 ± 0.088 n = 6 astrocytes, GABA: 0.26 ± 0.071 n = 9 astrocytes, *p* = 0.0093, unpaired Student’s *t*-test), which was again also the case when EGFP profiles were peak-scaled (glu: 0.47 ± 0.033 n = 6, GABA: 0.23 ± 0.060 n = 9, *p* = 0.0040, unpaired Student’s *t*-test). Thus, the average volume of astrocytic processes reached its plateau at glutamatergic synapses about 0.2 µm closer compared to GABAergic synapses.

Here, it needs to be considered if this finding is affected by the normalization we used. This is not the case because the EGFP intensity plateau at >0.8 µm represents distances at which the intensity of astrocytic EGFP and, thus, the amount of astrocyte volume has become uncorrelated with synapse position (Figure 3B). In fact, this plateau corresponds to the average fraction of tissue volume occupied by thin astrocyte processes of 5–10% in this brain region [20,30]. It also needs to be kept in mind that the analyzed region, a sphere with a radius of 1 µm, has a volume of ~4 µm^3^ and with an excitatory synapse density of ~2 µm^−3^ [40] it contains, on average, eight glutamatergic synapses, irrespective of whether the analysis was performed at a glutamatergic or GABAergic synapse. Therefore, no bias is introduced as data are being normalized to the EGFP fluorescence intensity corresponding to the average astrocyte volume fraction.

### 2.3. GABAergic Synapse Properties Correlate with Local Astrocytic Process Distribution

In the next set of analyses, we explored how the astrocytic volume distribution varied between GABAergic synapses with different properties. Because we wanted to compare astrocytic EGFP profiles between synapses with one or more gephyrin clusters, we centered the analysis on the center of mass of the VGAT immunolabeling (Figure 4A). First, we compared GABAergic synapses with one, two or three gephyrin clusters (Figure 4B). In the territory of each analyzed astrocyte, several VGAT clusters with one, two, or three gephyrin clusters were identified and EGFP profiles obtained, background corrected as above, and normalized to the average EGFP fluorescence recorded for synapses with one gephyrin cluster at a distance of 0.9–1.0 µm. No differences were observed.

We then asked if the astrocyte volume distribution depends on the size of the presynaptic bouton or the amount of postsynaptic gephyrin. For answering these questions, we averaged all astrocytic EGFP profiles independent of the number of gephyrin clusters to obtain the average EGFP distribution and a reference value (distance of 0.9–1.0 µm) for normalization per astrocyte. We then normalized the individual EGFP distributions per astrocyte to that reference and grouped them according to their integrated VGAT/gephyrin fluorescence intensity as low (below the median integrated VGAT/gephyrin fluorescence intensity) or high (above the median). EGFP profiles in both groups were then averaged per astrocyte (Figure 4C,D). Interestingly, there was more astrocytic volume at GABAergic synapses with strong adjacent gephyrin immunolabeling compared to those with lower gephyrin levels (Figure 4C), whereas the astrocytic volume distribution did not differ between synapses with low or high VGAT cluster size (Figure 4D). This suggests that the postsynaptic target or the postsynaptic properties of the GABAergic synapse control how strongly a GABAergic synapse is approached by astrocytic processes.

## 3. Discussion

Applying super-resolution expansion microscopy (ExM) to the visualization of GABAergic synapses, using VGAT as a presynaptic and gephyrin as a postsynaptic marker, and of nearby astrocytic processes enabled us to explore how the distribution of astrocytic processes depends on synapse properties. Our estimates of the GABAergic synapse density obtained using ExM are in the range of values reported previously, as explained above. Similar to previous studies [37,38,39], our results indicate that postsynaptic gephyrin clusters can be organized in subsynaptic domains. Such a distinct subsynaptic domain structure was not clearly detectable in our presynaptic VGAT labeling even though the fluorescence within a VGAT cluster was not always homogeneous (see for instance Figure 2B). Focusing on the number of clearly identifiable gephyrin clusters, we found that single VGAT clusters, presumably representing a single presynaptic GABAergic bouton, are in ~50% of the cases in contact with more than one postsynaptic gephyrin cluster. Although we have not visualized the postsynaptic cells themselves, it appears likely that these multiple postsynaptic gephyrin clusters contacting a single presynaptic VGAT cluster are on the same postsynaptic cell because the subsynaptic domain structure has also been observed on dendrites of cultured dissociated neurons [38,39]. Interestingly, there was no clear increase in the total presynaptic VGAT label and only a relatively small increase in the total gephyrin label at synapses with two or more gephyrin clusters. Because VGAT labels the presynaptic vesicle pool [31] and the gephyrin label correlates with postsynaptic receptor cluster size [37,39], this indicates that to some extent the number of postsynaptic subsynaptic domains and the size of the presynaptic vesicle pool and the total postsynaptic receptor pools are regulated independently.

We then investigated the distribution of astrocytic processes near GABAergic synapses and if and how it is related to synapse properties. As before [13,20], we quantified the astrocytic process volume at increasing distances from the synapse. The astrocytic process volume started to depart from zero at a distance of ~0.2 µm and reached near-plateau values at ~0.6 µm from the synapse. To obtain hints at the functional relevance of this finding, we compared this astrocyte volume distribution to that at glutamatergic synapses by re-analyzing a subset of previously published ExM data [13]. We found that astrocytic processes approach glutamatergic synapses more closely. A simple explanation of why astrocytic processes are further away from the center of the GABAergic synapse could be that GABAergic synapses are overall bigger than glutamatergic synapses, thus displacing more tissue volume and thereby astrocytic processes. However, the distance to half-maximum EGFP fluorescence (Figure 3C) is 0.44 ± 0.024 µm at GABAergic and 0.28 ± 0.020 µm at glutamatergic synapses (*p* < 0.001, unpaired Student’s *t*-test), indicating that the GABAergic synapse volume had to be about (0.44/0.28)^3^ = 3.9 times bigger than the glutamatergic in this scenario. Although total synapse size is difficult to quantify because glutamatergic synapses primarily contact spines and GABAergic do not [25], their presynaptic bouton volumes were estimated to be ~0.1 µm^3^ at glutamatergic synapses [41,42], which is similar to cortical inhibitory terminals [43], making this scenario unlikely. Additionally, a strong presynaptic VGAT labeling or more intense postsynaptic gephyrin signal should then be associated with a shift of astrocytic volume away from the synapse, which was however not the case (Figure 4C). Overall, these observations indicate that the astrocytic processes do not approach GABAergic synapses as closely as glutamatergic. 

This is interesting for several reasons. As pointed out above, the volume analyzed around a single GABAergic synapse contains on average about eight excitatory contacts. If, in this volume, excitatory contacts are more closely approached by astrocytic processes than GABAergic, then astrocytic processes need to selectively approach excitatory contacts while avoiding inhibitory. Thus, the distribution of astrocytic processes in the neuropil would not be random and growth would need to be directed and specific for the synapse type. However, GABAergic synapses are typically located on dendritic shafts, which occupy an unknown percentage of the perisynaptic volumes that we analyzed. Future studies using electron microscopy (EM) or other techniques that visualize all compartments together with a quantitative analysis as performed for glutamatergic synapses [14,15] could clarify this point. Previous studies using EM for the analysis of astrocytes at symmetric, and thus presumably inhibitory, synapses used a more qualitative approach by categorizing synapses on whether they were contacted by astrocytes at their clefts, pre- or postsynaptically, or not at all [12,24]. Although these studies did not explicitly compare the astrocyte surface or volume distributions between excitatory and inhibitory synapses, the reconstructed cell surfaces and volumes could be used to analyze and compare both. See, for instance, [14] for such an analysis. In either case, EM or ExM, the fixation protocol can influence experimental findings. Chemical fixation can, for instance, collapse extracellular space and alter perisynaptic astrocyte morphology measured using EM compared to cryo-fixation [44]. However, if tissue shrinkage and other fixation-induced alterations are homogeneous across the neuropil, then comparisons of volume distributions between synapses in the same sample, as carried out here, should be unaffected qualitatively. Further comparing the two approaches, ExM and EM, it should be noted that ExM, as carried out here, does not visualize the membrane and has a lower spatial resolution compared to EM. The advantages of ExM are its ability to visualize many proteins of interest (in this study EGFP, VGAT, GAT1, GAT3 and gephyrin) at super-resolution, the use of widely available confocal microscopy and the less time-consuming protocols and analyses, which allowed us to analyze relatively large tissue volumes and high cell and synapse numbers.

Irrespective of the underlying reason or mechanism, astrocytes approach GABAergic synapses less closely than glutamatergic. What would be a functional role of this synapse type-specific anatomical arrangement? One intriguing scenario is related to neurotransmitter uptake. The extracellular GABA concentration in the hippocampus is mainly controlled by the GABA transporters 1 and 3 (GAT1 and 3) [45]. The general view is that neuronal uptake of GABA is mainly mediated by GAT1, which is mostly expressed in presynaptic terminals of inhibitory neurons, whereas GAT3 is mostly responsible for astrocytic GABA uptake in the cortex [3,45,46]. As expected from the literature, ExM of GAT1 and 3 and visualization of EGFP-expressing astrocytes (Figure 5) confirmed that GAT3 is mostly located in astrocytes and especially on their membrane. This was not the case for GAT1, which was mostly found in axon-like structures. In contrast to GABA, glutamate released from synapses is believed to be mostly taken up by astrocytes [45,47]. Therefore, the different positioning of astrocytes near these synapse types could reflect the cellular distribution of neurotransmitter uptake: the presence of astrocytic processes very close to glutamatergic synapses could increase their efficiency in glutamate uptake and in limiting glutamate diffusion into extrasynaptic space. At GABAergic synapses, their relatively distant positioning to the synapse allows GAT1 in the presynaptic and axonal membrane to more efficiently re-uptake GABA by reducing the competition between neuronal GAT1 and the more distant astrocytic GAT3. For CA3-CA1 glutamatergic synapses, it has indeed been shown that the relative coverage of synapses by astrocytes determines the efficacy of local glutamate uptake [13] and that withdrawal of astrocytic processes from synapses after induction of synaptic long-term potentiation increases the escape of glutamate into extrasynaptic space and glutamatergic crosstalk [20]. For GABAergic synapses, such relationships between structure and structural plasticity and neurotransmitter diffusion and uptake remain to be established.

The latter could be of particular interest in the context of activity-dependent changes of GABAergic synapse structure [38,39,48,49]. In a recent study, Crosby and colleagues [39] demonstrated that pharmacological disinhibition of cultured neurons increases the total volume of postsynaptic gephyrin clusters and the number of subsynaptic gephyrin domains. Although we did not detect a dependency between the number of gephyrin clusters per synapse and the perisynaptic astrocytic volume distribution, astrocytic processes were more abundant at GABAergic synapses with a high total gephyrin labeling. This suggests that an activity-dependent growth or shrinkage of the GABAergic postsynaptic receptor pool could be associated with a structural remodeling of perisynaptic astrocytic processes. The heterogeneity of interneurons in the hippocampus [50] could be an alternative explanation of this specific finding. If the total amount of gephyrin at a GABAergic synapse depends on the cell type of the presynaptic and postsynaptic neuron, then the abundance of perisynaptic astrocytic processes could be controlled by the type of synaptic connection too.

Our observations reveal that astrocytic processes are scarcer at GABAergic compared to glutamatergic synapses. If the concept of the tripartite synapse would be defined exclusively by anatomy, then interactions between GABAergic synapses and perisynaptic astrocytic processes were less likely at this synapse type. However, functional interactions depend on the range of action of the involved signal or signaling molecules. At glutamatergic synapses, the directly measured action range of glutamate depends, for instance, on whether activation of AMPA or NMDA receptors is considered [51]. Such quantitative assessments that can be used as a reference for the functional interpretation of structural analyses of single synapses are at the moment mostly missing for GABAergic synapses.

## 4. Material and Methods

### 4.1. Animals

For this study, three male transgenic mice (FVB background) expressing EGFP under control of the human GFAP promotor [34] were sacrificed at an age of two to three months. Mice were bred in-house and reared under 12 h light/dark conditions with food and water ad libitum. All animal procedures were conducted in accordance with the regulations of the European Commission directive 2010/63/EU and all relevant national and institutional guidelines and requirements. All procedures have been approved by the Landesamt für Natur, Umwelt und Verbraucherschutz Nordrhein-Westfalen (LANUV, Germany) where required.

### 4.2. Expansion Microscopy

Expansion microscopy (ExM) was performed as described before [13,29] following published protocols [27,28,52]. After deep anesthesia (intraperitoneal injection of a mixture of ketamine 150 mg per kg of body weight and xylazine 15 mg per kg of body weight, injected volume 0.1 mL per 20 g of body weight) and subsequent intracardial perfusion of mice with 4% paraformaldehyde in phosphate-buffered saline (PBS, pH 7.4), brains were isolated and postfixed overnight at 4 °C. Brains were then washed three times with PBS and coronal hippocampal sections (70 µm thickness) were cut on a vibratome. For immunohistochemical labeling, sections were blocked overnight at 4 °C in blocking buffer consisting of 5% normal goat serum and 1% Triton X-100 in PBS. The sections were then incubated with primary antibodies in blocking buffer for 72 h at 4 °C. The following primary antibodies were used: chicken anti-GFP (1:2000; Abcam ab13970), mouse anti-Gephyrin (1:200; Synaptic Systems 147021), guinea pig anti-VGAT (1:200; Synaptic Systems 131004), rabbit anti-GAT1 (1:200; Synaptic Systems 274102), and guinea pig anti-GAT3 (1:200; Synaptic Systems 274304). Next, sections were washed 3 × 20 min in PBS at room temperature (RT) and incubated with secondary antibody in blocking buffer overnight at 4 °C. The following secondary antibodies were used: goat anti-chicken Alexa Fluor 488 (1:200; ThermoFisher A11039), goat anti-mouse Alexa Fluor 568 (1:200; ThermoFisher A11004), goat anti-guinea pig Alexa Fluor 633 (1:200; ThermoFisher A21105), goat anti-rabbit biotin (1:200; Jackson ImmunoResearch 111-066-144). After washing 3 × 20 min in PBS (RT), nuclei were stained with Hoechst 33,342 (1:2000 in distilled water, RT) for 10 min and sections were washed again 5 × 5 min in PBS (RT). The hemispheres of the brain sections were then separated and the hippocampus isolated. To later calculate the expansion factor, the nuclei staining of the tip of the dentate gyrus of each hemisection was imaged before expansion (see below for details). Next, slices were incubated with the linker methylacrylic acid-NHS (1 mM in PBS) for 1 h at RT and then washed for 3 × 20 min in PBS (RT). This was followed by an incubation for 45 min at 4 °C with monomer solution consisting of (in g/100 mL in PBS): 8.6 sodium acrylate, 2.5 acrylamide, 0.15 N,N-methylenebisacrylamide, 11.7 NaCl. Gelation of slices was achieved by incubating them for 5 min at 4 °C in gelling solution (monomer solution supplemented with 0.01% 4-hydroxy-TEMPO, 0.2% TEMED, 0.2% APS) followed by an incubation for 2 h at 37 °C in the same solution in a glass-bottom chamber covered with a coverslip. The coverslip was carefully removed, excess gel around the slice cut off, and the gels incubated overnight at 25 °C in digestion buffer consisting of: 50 mM Tris pH 8.0, 1 mM EDTA, 0.5% Triton-X 100, 0.8 M guanidine hydrochloride, 16 U/mL of proteinase K in distilled water. For staining with the rabbit anti-GAT1 antibody, gels were then incubated with Streptavidin-Alexa Fluor 647 (1:200; Jackson ImmunoResearch 016–600-084) in PBS for 1.5 h at RT. This was followed by the expansion in distilled water for 2–2.5 h at RT with water being exchanged every 10–20 min. Before imaging, expanded samples were mounted to poly-lysine (0.1% *w*/*v* in water)-coated µ-Slide 2-well Ibidi-chambers and covered with poly-lysine-coated coverslips. Deionized water was added at the sides to prevent the gel from drying.

Image stacks of EGFP-expressing astrocytes in the *stratum radiatum* of the hippocampal CA1 region were acquired on a Leica SP8 inverted confocal microscope using a 40x/1.1NA water immersion objective. Voxel dimensions were typically (x-y-z): 0.14 × 0.14 × 0.42 µm. Image stacks were then deconvolved using the Leica Systems Software. The expansion factor was determined by measuring the average diameter of ~10 Hoechst-stained nuclei at the tip of the dentate gyrus before and after expansion. On average, the expansion factor was 4.14 ± 0.07 (n = 11).

Scale bars in all illustrations of ExM and distances in analyses of ExM correspond to the pre-expansion size, i.e., the size/distance in the expanded specimen divided by the expansion factor obtained for that individual specimen.

### 4.3. Image Analysis

Image analysis of deconvolved images was performed with FIJI/ImageJ (NIH) and MATLAB (MathWorks). To determine the *cluster sizes and densities of gephyrin and VGAT*, the Analyze Particles Tool in FIJI was used. The image was binarized based on a custom threshold that was adjusted for the VGAT and gephyrin staining of each image stack individually to a value where background signals were removed while optically identified clusters were still visible. The minimum size filter for VGAT clusters was set to 0.04 µm^2^ [35]. For the gephyrin clusters, the size filter was estimated based on the size of the visually inspected clusters and set to 0.01 µm^2^. For calculation of the cluster sizes, clusters identified on the edges of the image were excluded while they were included for the calculation of the cluster densities. For each astrocyte z-stack, 6 sections at a z-distance of 4.2 µm were analyzed and the average cluster densities and sizes were calculated.

The quantification of the *number of gephyrin clusters per VGAT cluster* was performed by manual counting. For each astrocyte, 15 VGAT clusters were chosen randomly without inspecting the gephyrin signal. For each selected VGAT cluster, the number of gephyrin clusters was then determined visually.

Analysis of the *perisynaptic astrocytic volume distributions* was performed using custom-written macros in MATLAB similarly to the previously described analysis at excitatory synapses [13]. Within the territory of an EGFP-positive astrocyte, putative inhibitory synapses were selected based on close apposition of a presynaptic VGAT and a single postsynaptic gephyrin cluster. To avoid a selection bias, the EGFP channel was not visible during synapse identification. Starting from the manually selected putative center of the synaptic cleft, the astroglial EGFP fluorescence intensity was analyzed in spherical shells with a thickness of 20 nm starting with a radius of 0.020 µm and a final radius of 1 µm (Figure 3A). For each individual shell, the average of the EGFP fluorescence intensity was determined. For each of these EGFP-distance profiles, the background EGFP fluorescence was determined from the shells ≤ 0.080 µm (occupied by the GABAergic synapse) and subtracted. EGFP fluorescence profiles were obtained for 10 synapses for each astrocyte and averaged. To account for differences in EGFP expression between individual astrocytes and microscopy settings (e.g., excitation and fluorescence detection), the average EGFP fluorescence profile of each astrocyte was normalized to the average fluorescence in the last 100 nm in that profile (distance 0.9–1.0 µm). The normalized profiles per astrocyte were then averaged to obtain the final population average and SEM (Figure 3A,B). As a control, we performed the same analysis around randomly chosen locations in the image stack and used a volume occupied by a presynaptic terminal to determine the background EGFP fluorescence (Figure 3B).

For comparing the astrocytic volume distribution between inhibitory synapses with 1, 2 or 3 gephyrin clusters and low/high VGAT/gephyrin cluster sizes (Figure 4B–D), the analysis was slightly adapted. For the comparison between different gephyrin cluster numbers (Figure 4B), VGAT clusters with 1, 2 or 3 directly apposed gephyrin clusters (2–5 each) were randomly chosen per astrocyte without inspecting the EGFP signal. The center of mass of the presynaptic VGAT cluster was calculated based on fluorescence intensity and chosen as the center for the spherical analysis of EGFP fluorescence intensity, which was performed as described above. To account for differences in EGFP expression and detection between astrocytes, the EGFP profiles (1, 2, and 3 gephyrin clusters) were normalized to the average fluorescence intensity at a distance of 0.9–1.0 µm of the profiles with 1 gephyrin cluster for each astrocyte. Statistics are given for these averaged and normalized profiles (Figure 4B).

For the comparison of EGFP profiles between GABAergic synapses with low/high sizes of gephyrin/VGAT clusters (Figure 4C,D), the EGFP profiles at GABAergic synapses were categorized as low/high gephyrin/VGAT depending on whether the integrated gephyrin/VGAT fluorescence intensity of that GABAergic synapse was below/above the median of all integrated gephyrin/VGAT fluorescence intensities of the analyzed astrocyte. For each astrocyte, the average EGFP profiles were then calculated for the low and high groups and for all profiles. Then, for each astrocyte, the low/high profiles were normalized to the mean EGFP fluorescence intensity at a distance of 0.9–1.0 µm of all profiles. Statistics are given for these averaged and normalized profiles across all astrocytes (Figure 4C,D).

### 4.4. Statistical Analysis

Data calculations and statistical analyses were performed in MATLAB (MathWorks), Excel (Microsoft), GraphPad Prism (GraphPad Software), and Origin Pro (OriginLab Corporation). Results are given as mean ± standard error of the mean (SEM). The number of experiments (n) is specified for each data set. Statistical tests were used as stated in the figure legends and results section. Statistical significance is indicated by asterisks (* for *p* < 0.05, ** for *p* < 0.01 and *** for *p* < 0.001).

## Figures and Tables

**Figure 1 cells-11-03150-f001:**
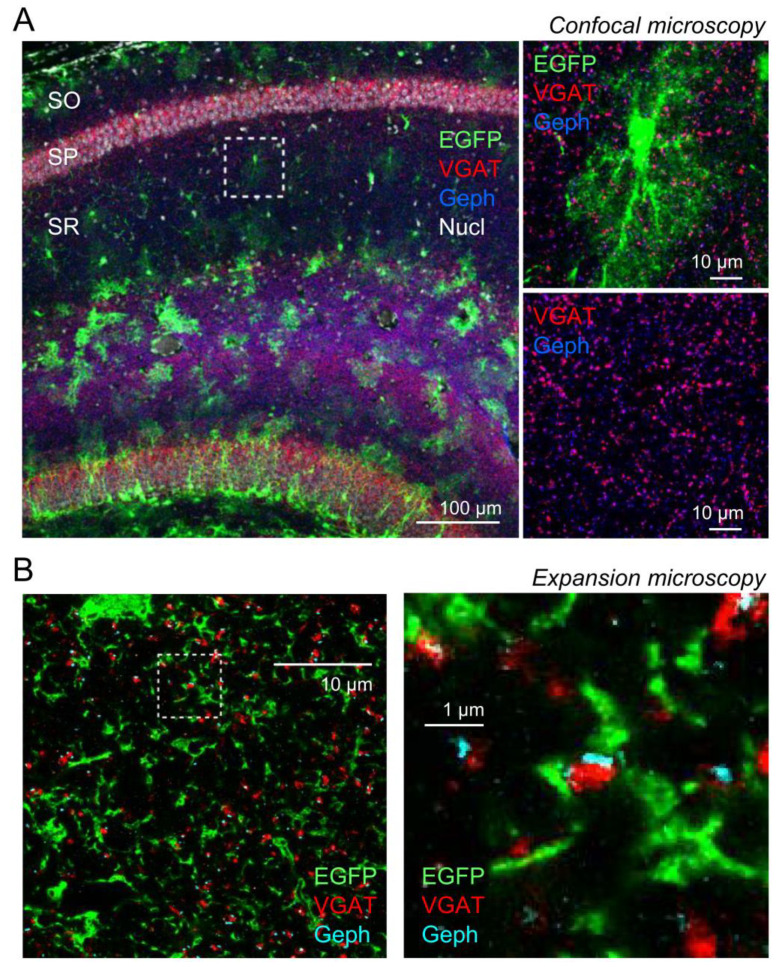
**High-resolution visualization of GABAergic synapses and perisynaptic astrocytic processes using expansion microscopy.** (**A**) Example of immunohistochemistry combining labeling of the vesicular GABA transporter (VGAT, red), the postsynaptic clustering protein gephyrin (Geph, blue), the EGFP expressed by astrocytes (green) and the nuclei (Nucl, Hoechst 33342, white). Confocal microscopy without expansion. Left panel: overview (SR: *stratum radiatum*, SO: *stratum oriens*, SP: *stratum pyramidale*). Right top panel: higher magnification of subregion outlined in the left panel (without Hoechst 33342). Right bottom panel: same subregion as above but only VGAT and gephyrin. (**B**) Example of the same immunohistochemical labeling after expansion obtained using the same confocal microscope (CA1, *stratum radiatum*). Left panel: overview of subsection of an astrocyte (cell body at the top border). Right panel: enlarged region outlined in the left panel. Note the sandwich-like structure of presynaptic VGAT and postsynaptic gephyrin in the center of the image.

**Figure 2 cells-11-03150-f002:**
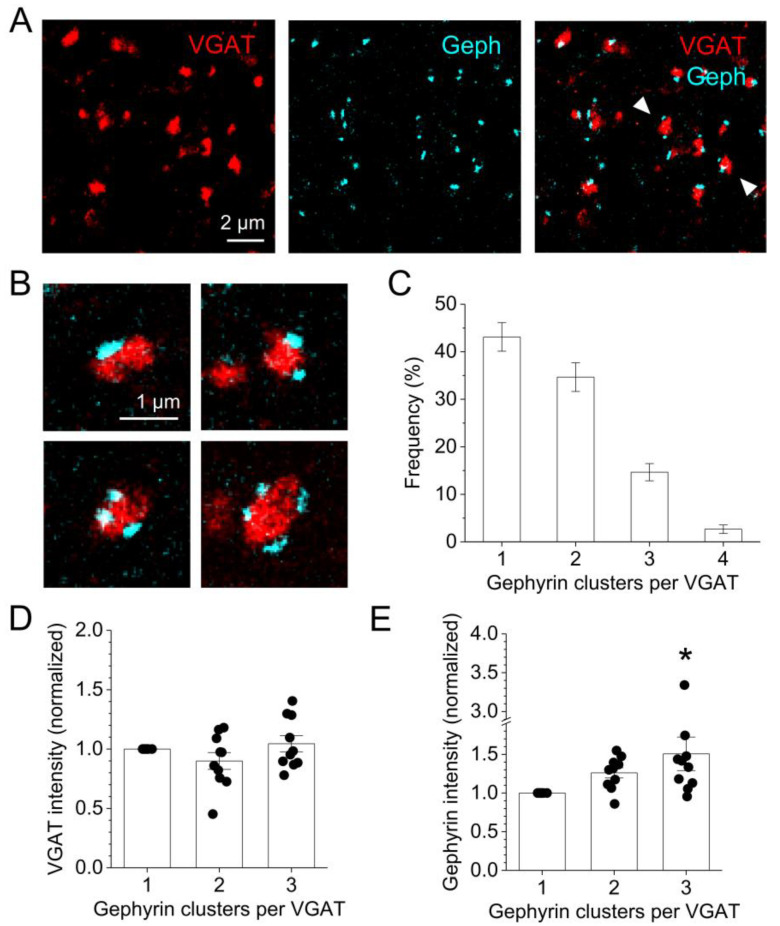
**Single GABAergic presynaptic terminals have multiple postsynaptic contacts.** (**A**) Example of post-expansion VGAT (left panel, red) and gephyrin (middle panel, cyan) immunolabeling. Right panel: overlay of VGAT and gephyrin reveals VGAT clusters surrounded by multiple gephyrin clusters (arrowheads). (**B**) Examples of VGAT clusters in contact with 1, 2, 3 or 4 gephyrin clusters. (**C**) Distribution of gephyrin cluster numbers per VGAT (total n = 225 VGAT clusters in the territory of 15 astrocytes from 5 independent experiments and 3 animals). (**D**) Analysis of integrated VGAT cluster fluorescence intensity contacting 1, 2, or 3 gephyrin clusters. Integrated VGAT fluorescence intensity was normalized per astrocyte to the value obtained for 1 gephyrin cluster (total n = 104 VGAT clusters in 10 astrocyte territories from 4 independent experiments). One-way ANOVA: *p* = 0.198. (**E**) Analysis of integrated gephyrin fluorescence intensity as in (**D**) (same data set). One-way ANOVA: *p* = 0.036. Tukey post hoc tests: 1 vs. 3 gephyrin clusters *p* = 0.028, *p* > 0.30 otherwise (without outlier near asterisk ANOVA *p* = 0.00225 and post hoc Tukey 1 vs. 3 clusters *p* = 0.00374). VGAT clusters contacted by 4 gephyrin clusters were not analyzed in (**D**,**E**) because there were too few per astrocyte territory for statistical analysis.

**Figure 3 cells-11-03150-f003:**
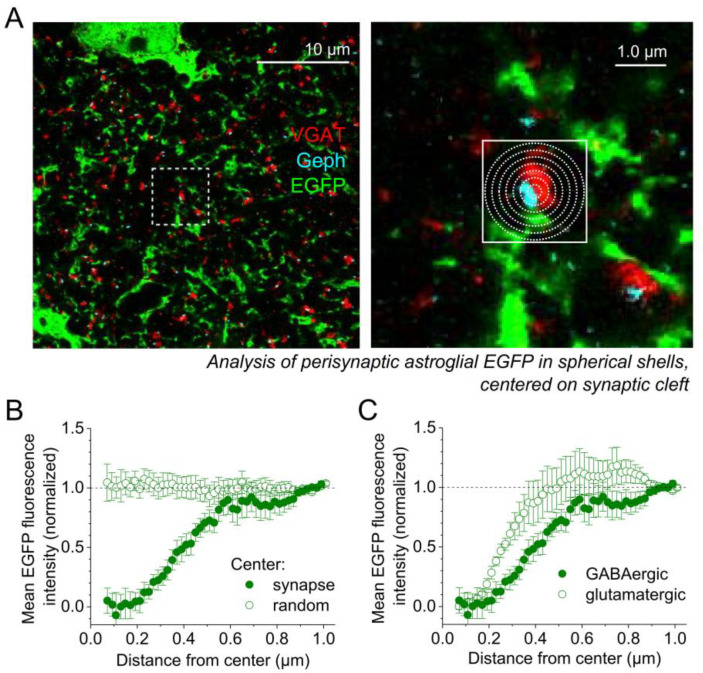
**Distribution of astrocyte volume at GABAergic synapses compared to glutamatergic synapses.** (**A**) Example of a visualization of an EGFP-expressing astrocyte (green), presynaptic GABAergic terminals (VGAT, red) and postsynaptic gephyrin clusters (Geph, cyan) using expansion microscopy. Left panel: overview. Right panel: zoomed in region outlined in the left panel. The astrocytic volume distribution was analyzed around GABAergic synapses with 1 gephyrin cluster by quantifying the average EGFP fluorescence intensity in concentric three-dimensional shells with increasing diameter centered on the middle of the assumed synaptic cleft between the VGAT and gephyrin cluster (not all shells shown). EGFP fluorescence profiles were averaged and normalized per astrocyte (see main text and methods). (**B**) EGFP fluorescence increases as a function of distance and reaches near maximum levels at 0.6–1.0 µm from the synapse (filled circles: analysis centered on synapses, empty circles: random location). Repeated measures one-way ANOVA of synapse-centered analysis: distance *p* < 0.001, n = 9 average profiles (86 synapses in 9 astrocyte territories from 3 independent expansion microscopy experiments, 3 animals, filled circles). Analysis of 100 randomly placed volumes of interest from the same data set (empty circles, repeated measures one-way ANOVA *p* = 1.0). (**C**) Comparison of astrocytic EGFP distribution at GABAergic synapses (same data as in (**B**) with a re-analysis of the EGFP distribution at glutamatergic synapses from [13] (121 synapses in 6 astrocyte territories from 3 independent expansion microscopy experiments). Repeated measures two-way ANOVA: distance *p* < 0.001, interaction of distance and synapse type *p* = 0.0128.

**Figure 4 cells-11-03150-f004:**
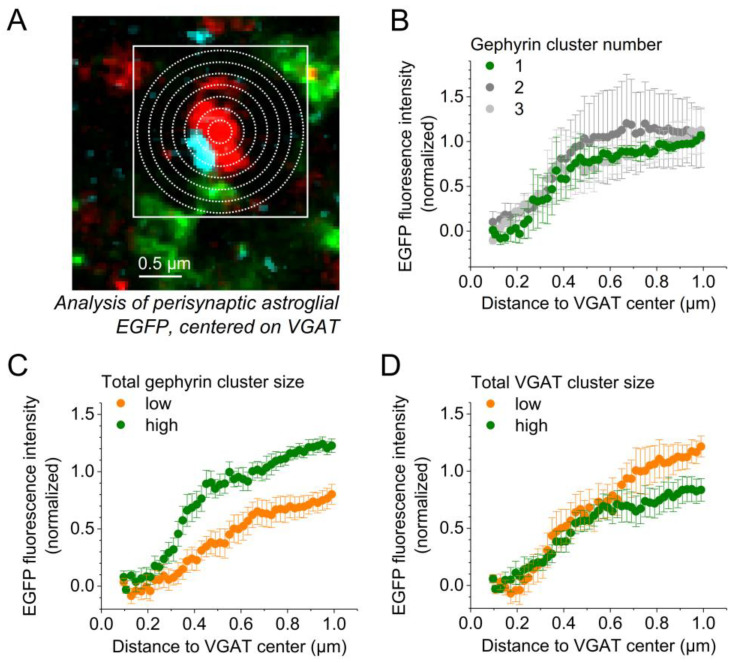
**Synapse-specific distribution of astrocytic processes around GABAergic synapses.** Additional analyses of experimental data obtained for Figure 3. (**A**) The average astrocytic EGFP fluorescence intensity (green) was determined in concentric shells with increasing diameter (white circles) centered on the center of mass of the presynaptic VGAT fluorescence (red) to quantify the astrocyte volume distribution around GABAergic synapses and its dependence on distance from the synapse (postsynaptic gephyrin, cyan). (**B**) For comparison of astrocyte volume profiles (EGFP fluorescence intensity) around GABAergic synapses with different numbers of gephyrin clusters, EGFP profiles were grouped according to the number of gephyrin clusters, averaged and normalized. Repeated measures two-way ANOVA: distance *p* < 0.001, interaction of distance and gephyrin cluster number *p* > 0.999, n = 6 average profiles from 1, 2, and 3 gephyrin clusters. Analysis of 57 synapses from 6 astrocytes from 3 independent expansion microscopy experiments (3 animals). (**C**) Comparison of astrocyte volume profiles (EGFP fluorescence intensity) around GABAergic synapses with small and large postsynaptic gephyrin clusters. For each astrocyte, EGFP profiles were grouped according to the gephyrin cluster fluorescence intensity (low = below the median intensity of gephyrin clusters of that astrocyte, high = above that median), averaged and normalized. Repeated measures two-way ANOVA: distance *p* < 0.001, interaction of distance and high/low *p* < 0.001, n = 9 average profiles for high and low. Analysis of 86 synapses from 9 astrocytes from 3 independent expansion microscopy experiments (3 animals). (**D**) Comparison of astrocyte volume profiles (EGFP fluorescence intensity) around GABAergic synapses with small and large presynaptic VGAT clusters. Analysis as in (**C**). Repeated measures two-way ANOVA: distance *p* < 0.001, interaction of distance and high/low *p* > 0.999, n of synapses, astrocytes, and experiments as in (**C**).

**Figure 5 cells-11-03150-f005:**
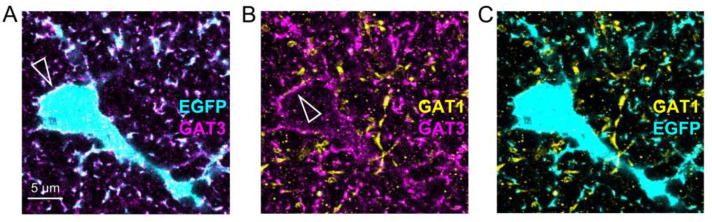
**Colocalization of GABA transporters 3 (GAT3) but not 1 (GAT1) with astrocytes.** Example of an expansion microscopy experiment of an astrocyte expressing EGFP (cyan) with GAT1 (yellow) and GAT3 (magenta). (**A**) Colocalization of EGFP and GAT3. Note the GAT3 label at the astrocyte cell surface (e.g., at the cell body, white triangle). (**B**) Lack of colocalization between GAT1 (yellow) and GAT3 (magenta). Note how GAT3 outlines the astrocyte surface (white triangle). (**C**) Lack of colocalization between astrocytic EGFP (cyan) and GAT1 (yellow). Scale bar in (**A**) applies to all panels.

## Data Availability

The datasets supporting the current study have not been deposited in a public repository but are available upon reasonable request from the corresponding author.

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
