# Peer review of "Structural Heterogeneity of the GABAergic Tripartite Synapse"

_cells, 2022, doi:10.3390/cells11193150_

Round 1

Reviewer 1 Report

Brunskine and colleagues have here presented important morphological work pertaining the anatomical interaction between astrocytic processes and glutamatergic and GABAergic synapses by super resolution expansion microscopy. The rationale for the study is well presented, and the analyses and conclusions seems justified. I have three minor comments:

1) For expansion microscopy, the tissue has undergone extensive processing. In particular, perfusion, that we do not know for sure how will preserve the ultrastructure of the tissue. Astrocytic processes are known to be motile and also change their morphology due to insults (for instance when preparing brain slices) and likely change dynamically in physiology and potentially due to for instance anesthesia. I think the discussion should include some comments about these caveats in the interpretation of the data.

2) As far as I can read the specific anesthesia used before perfusion is not stated, but should be included.

3) In figure 2E bar for 3 gephyrin clusters, there seems to be a clear outlier at about twice the maximum intensity as the rest of the observations. Was there anything in particular with this particular synapse? Given parametric statistics and a p-value close to 0.05 - what would be the effect of removing this outlier on differences between the group? It would be nice to know that the conclusion of a modest increase is not driven by one extreme outlier. 

Author Response

Thank you very much for the helpful comments. We have revised the manuscript accordingly. Changes in the text are in red.

[…]

1) For expansion microscopy, the tissue has undergone extensive processing. In particular, perfusion, that we do not know for sure how will preserve the ultrastructure of the tissue. Astrocytic processes are known to be motile and also change their morphology due to insults (for instance when preparing brain slices) and likely change dynamically in physiology and potentially due to for instance anesthesia. I think the discussion should include some comments about these caveats in the interpretation of the data.

We expanded the discussion to address this point. The study by Korogod et al. (2015, eLife) directly investigated the role of tissue fixation and is now cited. If tissue changes introduced by fixation are homogeneous throughout the neuropil, they should not affect the qualitative outcome of our volume distribution comparisons within the same sample. But it is indeed correct that we cannot provide direct experimental evidence for that at this time.

2) As far as I can read the specific anesthesia used before perfusion is not stated, but should be included.

Mice were anesthetized by intraperitoneal injection of a mixture of ketamine 150 mg per kg of body weight and xylazine 15 mg per kg of body weight (injected volume 0.1 ml per 20 g of body weight). This is now stated in the manuscript.

3) In figure 2E bar for 3 gephyrin clusters, there seems to be a clear outlier at about twice the maximum intensity as the rest of the observations. Was there anything in particular with this particular synapse? Given parametric statistics and a p-value close to 0.05 - what would be the effect of removing this outlier on differences between the group? It would be nice to know that the conclusion of a modest increase is not driven by one extreme outlier.

We had noticed this outlier before but did not find anything unusual in the raw data or the analyses from this astrocyte. Removing this outlier reduces the p-value of the ANOVA to 0.00225 and of the comparison between 1 and 3 clusters to 0.00374. Thus, the conclusions are not affected by this outlier. We have added the additional statistical analyses to the figure legend.

Reviewer 2 Report

The authors used super-resolution expansion microscopy to unravel the organization of astrocytic processes at GABAergic synapses. This is an interesting question as in contrast to the situation at excitatory synapses far less is known about the contact sites between inhibitory synapses and astrocytes. Therefore, I consider this a timely topic.

The manuscript is well written and the organization of figures and main text is clear and logical. The statistics used are adequate. The authors analyzed eGFP fluorescence intensity in the vicinity of GABAergic synapses taking advantage of super-resolution expansion microscopy. I agree also that working with surface reconstruction is very sensitive to how the thresholds are adjusted. Therefore the authors used instead assessment of eGFP signal intensity as previously established in the group and normalized to compensate for differences in eGFP expression between astrocytes. A direct comparison between excitatory and inhibitory synapses revealed that perisynaptic astroglial processes are more enriched at glutamatergic terminals. In Figure 3, the same data set is shown in B and C which I find redundant as the eGFP fluorescence curves can also be studied clearly in C and directly compared to each other. Whereas the fluorescence profile obtained from glutamatergic synapses clearly shows a plateau it seems less obvious if the GABAergic terminals are even specifically targeted by astrocytic processes in this analysis as the plateau is less obvious. The authors should comment on this. Maybe it would be possible to control for this by for instance randomly selecting non-synaptic center points and assess the eGFP profile as the authors also mentioned correctly that 5-10% of tissue in this brain region are occupied by astrocytic processes.

Author Response

We thank the reviewer for the helpful comments. We have performed additional analyses and changed the manuscript accordingly. Changes in the text are in red.

[…] In Figure 3, the same data set is shown in B and C which I find redundant as the eGFP fluorescence curves can also be studied clearly in C and directly compared to each other. Whereas the fluorescence profile obtained from glutamatergic synapses clearly shows a plateau it seems less obvious if the GABAergic terminals are even specifically targeted by astrocytic processes in this analysis as the plateau is less obvious. The authors should comment on this. Maybe it would be possible to control for this by for instance randomly selecting non-synaptic center points and assess the eGFP profile as the authors also mentioned correctly that 5-10% of tissue in this brain region are occupied by astrocytic processes.

We have performed an additional analysis, in which we calculated the volume distribution of astrocytic processes around randomly chosen locations in the same image stacks. This volume distribution shows no distance dependency. We have added this analysis to Fig. 3B, which is indeed more useful now.

Reviewer 3 Report

This paper uses super-resolution expansion microscopy to characterize the structural relationship between astroglial processes and GABAergic synapses in stratum radiatum of the mouse hippocampal CA1 region.  This is a welcome piece of work to add to the relatively less studied field of inhibitory synapses and their structural relationship to astroglial processes.

The paper is well-written and the data are presented clearly.  There are only some minor suggestions:

In abstract,

Line 11, consider change the wording “very little” to  (less)

Line 19 , consider change “unexpected” to  (novel)

In discussion,

Line 274, why “surprisingly”?  When in line 53 the authors suspect that “the astrocytic processes near excitatory and inhibitory synapses is likely to differ”

Line 303. Consider change the wording “did not explicitly compare excitatory and inhibitory synapses [12,24].”  Both of these EM references did considerable analysis on excitatory and inhibitory synapses as stated in line 302.  Indeed, the parameters studied in these EM studies were different from the “distance analysis” carried out here, and a bit more discussion would be appreciated.  For example, how would a comparable “distance analysis” be carried out with the 3D EM data?

In view of the previous EM 3D studies (12, 24, 36), I would like to see more discussion on the advantages of the authors current approach over EM analysis, for example, perhaps a larger volume of brain regions can by analyzed in a shorter time.

Author Response

Thank you very much for your helpful comments. We have revised the manuscript accordingly. Changes in the text are in red.

[…]

Line 11, consider change the wording “very little” to  (less)

Done.

Line 19, consider change “unexpected” to  (novel)

Done.

Line 274, why “surprisingly”?  When in line 53 the authors suspect that “the astrocytic processes near excitatory and inhibitory synapses is likely to differ”

We removed the word “surprisingly”.

Line 303. Consider change the wording “did not explicitly compare excitatory and inhibitory synapses [12,24].” Both of these EM references did considerable analysis on excitatory and inhibitory synapses as stated in line 302.  Indeed, the parameters studied in these EM studies were different from the “distance analysis” carried out here, and a bit more discussion would be appreciated.  For example, how would a comparable “distance analysis” be carried out with the 3D EM data?

We have rephrased this section and state that further analyses of cell surfaces from EM reconstructions could be helpful (page 9 of the revised manuscript).

In view of the previous EM 3D studies (12, 24, 36), I would like to see more discussion on the advantages of the authors current approach over EM analysis, for example, perhaps a larger volume of brain regions can by analyzed in a shorter time.

We have expanded this section of the discussion and mention some of the relevant advantages and disadvantages of ExM compared to EM (page 9 of the revised manuscript).